# Low Expression of Mitofusin 1 Gene Leads to Mitochondrial Dysfunction and Embryonic Genome Activation Failure in Ovine-Bovine Inter-Species Cloned Embryos

**DOI:** 10.3390/ijms231710145

**Published:** 2022-09-04

**Authors:** Shanshan Wu, Xiaoyu Zhao, Meiling Wu, Lei Yang, Xuefei Liu, Danyi Li, Han Xu, Yuefang Zhao, Xiaohu Su, Zhuying Wei, Chunling Bai, Guanghua Su, Guangpeng Li

**Affiliations:** 1State Key Laboratory of Reproductive Regulation and Breeding of Grassland Livestock (R2BGL), Inner Mongolia University, 24 Zhaojun Rd., Hohhot 010070, China; 2College of Life Sciences, Inner Mongolia University, 24 Zhaojun Rd., Hohhot 010070, China

**Keywords:** mitochondrial fusion, embryonic genome activation, transcriptomic sequencing, interspecies somatic cell nuclear transfer

## Abstract

Inter-species somatic cell nuclear transfer (iSCNT) is significant in the study of biological problems such as embryonic genome activation and the mitochondrial function of embryos. Here, we used iSCNT as a model to determine whether abnormal embryo genome activation was caused by mitochondrial dysfunction. First, we found the ovine-bovine iSCNT embryos were developmentally blocked at the 8-cell stage. The reactive oxygen species level, mitochondrial membrane potential, and ATP level in ovine-bovine cloned embryos were significantly different from both bovine-bovine and IVF 8-cell stage embryos. RNA sequencing and q-PCR analysis revealed that mitochondrial transport, mitochondrial translational initiation, mitochondrial large ribosomal subunit, and mitochondrial outer membrane genes were abnormally expressed in the ovine-bovine embryos, and the mitochondrial outer membrane and mitochondrial ribosome large subunit genes, mitochondrial fusion gene 1, and ATPase Na+/K+ transporting subunit beta 3 gene were expressed at lower levels in the ovine-bovine cloned embryos. Furthermore, we found that overexpression and knockdown of *Mfn1* significantly affected mitochondrial fusion and subsequent biological functions such as production of ATP, mitochondrial membrane potential, reactive oxygen species and gene expressions in cloned embryos. These findings enhance our understanding of the mechanism by which the *Mfn1* gene regulates embryonic development and embryonic genome activation events.

## 1. Introduction

Terminal differentiated cells can be reprogrammed to a totipotency state through somatic cell nuclear transfer (SCNT), which is widely used in animal breeding and gene-edited animals for biomedical research [1,2]. In interspecific somatic cell nuclear transfer (iSCNT), donor cells are transplanted into a recipient enucleated oocyte of a different species/family/order/class to construct a heterogeneous reconstructed embryo [3]. iSCNT technology is potentially significant for rescuing endangered animals and therapeutic cloning [4]. However, the remolded SCNT or iSCNT embryos have often been blocked in the early stages of development when the reconstructed embryo fails to activate embryonic genome activation (EGA), which occurs at the 2-cell stage in mice, 4- to 8-cell stages in humans, and 8- to 16-cell stages in cattle [5,6]. Many studies have found abnormalities such as chromosomal aberrations, incompatibility between mitochondrial and nuclear genomes, large offspring syndrome, and placental dysfunction-related disorders, which often occur in cloned embryos, conceptuses, and offspring [1,7,8,9], and have attributed these to incomplete or unfaithful somatic cell nuclear reprogramming (NR) [2,10]. These abnormalities might be the result of incorrect NR, which ultimately leads to improper mitochondrial genome activation and mitochondrial malfunctions [11,12]. Mitochondrial dysfunction may occur after nuclear transfer due to a failure of nuclear remodeling by failed nucleus-cytoplasmic interaction.

The mitochondria has a wide range of functions beyond ATP synthesis, including the generation of intracellular reactive oxygen species and regulation of calcium, and actively participates in the regulation of signal transduction pathways [13,14,15]. Critical nuclear and cytoplasmic interactions may be determined by mitochondria. There is a great deal of communication between nuclear and mitochondrial genomes. Mitochondria synthesize ATP, an important energy currency, in many processes including oocyte maturation, fertilization, and early embryonic development [16,17,18]. Mitochondrial dysfunction may lead to the activation of apoptosis cascades that result in abnormal embryo development and defects [19,20,21].

In mammals, the mitochondria are remarkably dynamic organelles, and their morphology is maintained by a balance between fusion and fission. When fusion is suppressed, small and fragmented mitochondria can be observed. When fission is suppressed, slender, interconnected mitochondrial networks are observed. In order to maintain mitochondrial quality and cellular function, the control of the size, position, and shape of mitochondria within a cell requires a balance between fusion and fission [22,23]. Mitochondrial fusion has been reported to be essential for embryonic development and, by enabling cooperation between mitochondria, exerts protective effects on the mitochondrial population [24]. Mitofusin1 and mitofusin2 (*Mfn1* and *Mfn2*), the homologues of *Fzo* in yeast and *Drosophila*, are the critical regulators of mitochondrial fusion in mammalian cells [25,26,27]. Mice with deletions of *Mfn1* or *Mfn2* died mid-gestation [24]. Embryonic fibroblasts lacking *Mfn1* and *Mfn2* have defects in mitochondrial membrane potential and respiratory capacity [28]. Mutations in *Mfn2* caused neurodegenerative diseases by disrupting mitochondrial organization [29]. Knockdown of *Mfn2* in human iPSCs has been reported to result in defects in neurogenesis and synapse formation because of disruptions in mitochondrial metabolism and the mitochondrial network [30]. Moreover, mice lacking the fission protein DRP1 exhibited developmental abnormalities and died after embryonic day 12.5, but the mechanism was not identified [31]. In our previous study, we found that the major difference in transcriptome of 8-cell embryos between SCNT and iSCNT embryos was merely enriched in mitochondrial fusion genes and mitochondrial outer membrane protein genes, which implied that mitochondrial genes may affect the EGA of iSCNT embryos [32].

These studies indicate that the abnormalities in the early nuclear transfer embryos may result in blocked cloned embryo development, but there are few studies that focus on the genes related to mitochondrial function, especially those related to mitochondrial membrane and mitochondrial fusion. In this study, we analyzed the transcriptome of SCNT and iSCNT embryos, focusing on the mitochondrial fusion genes and the mechanism of EGA in cloned embryos. This study provides comprehensive insights into the effects of mitochondrial fusion genes on the reprogramming and efficiency of bovine SCNT embryos.

## 2. Results

### 2.1. Abnormal Mitochondrial Function Led to the Arrest of O-B Inter-Species SCNT Embryos in 8-Cell Stage

In order to study the development efficiency of different embryos, we compared the in vitro development efficiency of bovine-bovine (B-B) and ovine-bovine (O-B) embryos with bovine IVF embryos as the control. The results showed that there was no significant difference in the development rates of IVF, B-B and O-B embryos at the 2-cell stage (79.96%, 78.41% and 74.46%, respectively) and 8-cell stage (75.45%, 72.05% and 66.96%, respectively). However, the developmental efficiency of O-B embryos was significantly reduced in morula and blastocyst stages (*p* < 0.001), indicating that the development of O-B embryos was blocked at the 8–16 cell stage (Figure 1a,b, Appendix A). We then measured the ROS levels of the embryos at the 8-cell stage. Through analysis, we found that IVF embryos had the lowest ROS level and the ROS levels in B-B embryos and O-B embryos were significantly higher than in IVF embryos (*p* < 0.001, Figure 1c,d). Mitotracker staining showed that the concentration of mitochondria in B-B and O-B embryos was significantly lower than that in the IVF group (*p* < 0.05); moreover, there was a greater concentration of mitochondria in B-B embryos compared with O-B embryos (Figure 1f,g). Membrane potential JC-1 staining showed that the membrane potential ratio of the IVF group and the B-B group was significantly higher than that in the O-B group (*p* < 0.001, Figure 1h,i). ATP content in O-B embryos was significantly lower than those in B-B and IVF embryos (*p* < 0.001, Figure 1e).

### 2.2. Transcriptome Sequencing Analysis Showed That Gene Expression Was Abnormal in O-B Embryos at 4- to 8-Cell Stage

Since O-B embryos have blocked development at the 8-cell stage, we collected 4-cell, 8-cell and 16-cell embryos of IVF, B-B and O-B embryos, respectively, for RNA-seq sequencing to understand the causes of impaired development (Figure 2a). PCA analysis showed that 4-, 8- and 16-cell of O-B embryos were clustered with the 4-cell stage of IVF and B-B embryos, while 8-cell of IVF embryos were clustered with B-B 8-cell embryos, and IVF 16-cell embryos were clustered with 16-cell B-B embryos (Figure 2b). Sample correlation analysis showed that the correlation between O-B 8-cell embryos and IVF and B-B 8-cell embryos is 0.448 and 0.381, respectively, which was significantly lower than that between IVF and B-B embryos of 0.929 (Figure 2c). Heat map analysis of differential gene expression (DEGs) patterns revealed that the gene expression patterns of O-B at 4-, 8- and 16-cells are clustered with the expression patterns of 4-cell IVF and B-B embryos, while the expression patterns of 8-cells and 16-cell stage of IVF and B-B embryos are clustered together (Figure 2d). These results indicated that O-B embryos did not initiate EGA at the 8-cell stage.

### 2.3. Mitochondrial Function Related Genes and Pluripotency Genes Were Not Activated in O-B Embryos

The above transcriptomics results show that O-B embryos have obvious abnormalities in gene expressions at the 8-cell stage. Therefore, we further compared and analyzed the 8-cell stage embryos from IVF, B-B, and O-B embryos. We divided gene expression between different source embryos into four groups, named Groups I to IV. Group I includes specific target genes in both IVF and B-B embryos (1472 genes); Group II includes genes expressed only in O-B embryos (1826 genes); Group III includes genes expressed only in B-B embryos (876 genes); and Group IV includes genes only expressed in IVF embryos (762 genes) (Figure 3a). Heat map analysis of the 118 mitochondrial-related genes in Group I showed that IVF-8-cell and B-B-8 cell embryos were clustered together, IVF and B-B 16-cell embryos were clustered together, and O-B 4-, 8-, 16-cell embryos were clustered together with IVF 4-cell and B-B 4-cell embryos (Figure 3b). These results suggest that the mitochondrial-related genes in O-B embryos have not been activated. GO enrichment analysis of the 118 genes indicated that these genes were concentrated in the mitochondrial transport, mitochondrial translational initiation, mitochondrial large ribosomal subunit, mitochondrial outer membrane and other pathways (Figure 3c). In addition, we also found that bovine ZGA-related pluripotency genes, including *Klf4*, *Otx1*, *Kdm5b*, *Tet1*, *Taf1d*, *Sp1*, *Nfya*, *Nfyb*, *Dppa4* and *Nanog* in OB embryos were significantly lower than those in IVF and B-B embryos. Surprisingly, the *Pou5f1* gene was significantly upregulated in O-B embryos (Figure 3d).

### 2.4. Expression Analysis of Mitochondrial Function Related Genes in IVF, B-B and O-B 8-Cell Stage Embryos

Since there were significant differences in the expression of genes related to mitochondrial function in O-B iSCNT 8-cell stage embryos compared to B-B SCNT and IVF 8-cell stage embryos, we selected genes related to mitochondrial function for further study, including ATPase Na+/K+ transporting subunit beta 3 gene (*Atp1b3*), mitochondrial outer membrane gene (*Tomm40*), and mitochondrial ribosome large subunit gene (*Mrpl48*). We found that the expression trend of *Atp1b3* gene mRNA was consistent in RNA-SEQ and Q-PCR data (Figure 4a,b). The expression of ATP1B3 protein in O-B embryos was significantly lower than that in B-B embryos (*p* < 0.05), and significantly lower than that in IVF embryos (*p* < 0.01). Immunofluorescence analysis of the embryos showed that ATP1B3 proteins in O-B embryos and B-B embryos were expressed at significantly lower levels than that in IVF embryos (Figure 4c,d) (*p* < 0.05). RNA SEQ and Q-PCR data indicated that the expression of *Tomm40* gene mRNA in O-B embryos was significantly lower than in B-B embryos (*p* < 0.01 and IVF embryos (*p* < 0.001) (Figure 4e,f). The protein level of TOMM40 in O-B and B-B embryos was significantly higher than in IVF embryos (*p* < 0.05) (Figure 4g,h). The expression of *Mrpl48* gene mRNA in O-B embryos was significantly lower than in B-B embryos (*p* < 0.05) and IVF embryos (*p* < 0.01) (Figure 4i,j). Immunofluorescence staining showed that the expression levels of MRPL48 proteins in O-B embryos and B-B embryos were significantly higher than in IVF embryos (*p* < 0.05) (Figure 4k,l). These results suggest that the abnormal expression of mitochondrial-related genes in O-B embryos may be one of the main causes of the developmental block and failure of O-B embryos.

### 2.5. Mitochondrial Fusion Gene Mfn1 Affected Embryonic Development and Improved Embryonic Mitochondrial Metabolism

Our previous study found that mitochondrial fusion was significantly enriched in iSCNT embryos [32]. To further investigate the effects of mitochondrial function on embryonic development, we focused on mitochondrial fusion proteins. By analyzing RNA-SEQ data of mitochondrial fusion proteins MFN1 and MFN2 in IVF, B-B and O-B 8-cell stages embryos, we found that the expression of *Mfn1* and *Mfn2* genes in O-B cells was lower than in IVF and B-B embryos at the 8-cell stage (Figure 5a). We found similar results using Q-PCR for verification (Figure 5b).

To further test the effect of *Mfn1* on SCNT embryos, we first constructed *Mfn1* overexpression and knockdown vectors. The *Mfn1* overexpression vector contained CMV promoter and kanamycin resistance gene tag (Figure 5c). After the transfection of fetal fibroblasts, we selected a cell line (OE-Mfn1 cells) that stably overexpressed the *Mfn1* gene for further study. We found *Mfn1* mRNA and protein expression were significantly increased in the OE-Mfn1 cells (Figure 5d–f). In addition, we designed, synthesized, and tested four Mfn1 shRNA vectors (shRNA1, shRNA2, shRNA3 and shRNA4) (Figure 5g). After the transfection of fetal fibroblasts, we found shRNA1 had the best knockdown efficiency of both mRNA and protein in cells (named the shMfn1) (Figure 5h–j). We further used the OE-Mfn1 cells and the shMfn1 cells as donor cells to construct the cloned embryos and evaluated the *Mfn1* expressional verification test in the B-B and O-B 8-cell stage embryos. There was significantly more MFN1 protein in B-B OE-MFN1 and O-B OE-MFN1 8-cell stage embryos compared to B-B and O-B cloned embryos. Moreover, MFN1 fluorescence intensity in B-B OE-MFN1 embryos was similar to the intensity in IVF embryos. However, after knockdown *Mfn1* gene, MFN1 protein signals in B-B shMFN1 and O-B sh-MFN18-cell stage embryos decreased (Figure 5k).

To evaluate the effect of the *Mfn1* gene specifically on embryo development, we used OE-Mfn1 and shMfn1 cell lines to construct SCNT embryos, and we used B-B and O-B cloned embryos as controls. We found that the overexpression of *Mfn1* improved 8-cell and blastocyst development in both B-B and O-B OE-Mfn1 embryos. However, knockdown of *Mfn1* significantly reduced 8-cell and blastocyst rates in B-B and O-B shMfn1 embryos (Figure 6a,b, Appendix A). Examination of ROS level in 8-cell embryos showed that the ROS level in B-B and O-B OE-Mfn1 8-cell embryos were significantly lower than in B-B and O-B embryos and higher than in O-B shMfn1 embryos (Figure 6c). Mitotracker staining showed that the density of mitochondria in B-B and O-B OE-Mnf1 was significantly higher than in B-B and O-B embryos, but with low expression of O-B shMfn1 (Figure 6d,e). Analyses of ATP level and mitochondrial membrane potential indicated that overexpressed *Mfn1* increased ATP level and mitochondrial membrane potential in B-B and O-B OE-Mfn1 8-cell embryos. However, knockdown of *Mfn1* decreased ATP and mitochondrial membrane potential in B-B and O-B shMfn1 8-cell embryos (Figure 6f,g).

We conducted RT-qPCR validation of EGA-related genes to explore the effect of overexpression and knockdown of *Mfn1* on embryonic EGA. We found that overexpression of the *Mfn1* gene could partly correct the EGA-related pluripotency gene expression, such as *Klf4*, *Otx1*, *KDM5B*, *Tet1*, *Sp1*, *Nfya*, *Nfyb*, *Dppa4* and *Nanog* in B-B OE-Mfn1 and O-B OE-Mfn1 8-cell embryos, which were similar to those in IVF embryos. However, in B-B shMfn1 embryos, the *Pou5f1*, *Tfap2c* and *Taf1d* genes expression were significantly up-regulated compared with IVF embryos (Figure 6h).

## 3. Discussion

Somatic cell nuclear transfer is assisted reproductive technology that can reprogram a differentiated donor cell into a totipotent state [33]. Interspecies somatic nuclear transfer provides extreme cases of reprogramming failure and can be used to understand the basic biological mechanisms of genome reprogramming. It also provides an important tool for studying the interaction between the nucleus and cytoplasm of embryos before implantation. The coordination and interaction between nucleus and cytoplasm is the prerequisite of nuclear reprogramming and guides the development process [34]. Mitochondria are the most abundant organelles in the oocyte cytoplasm, and their structure and distribution change significantly during oocyte maturation and embryonic development. Furthermore, mitochondria, as ATP-producing organelles, play an important role in oocyte maturation and embryo development [24]. In this study, we found that the developmental block in O-B cloned embryos is caused by abnormal mitochondrial function. The mitochondrial distribution and membrane potential in O-B cloned 8-cell embryos were obviously abnormal, and the ATP concentration was lower than that of IVF embryos and B-B cloned embryos. Moreover, ROS levels in O-B embryos were higher than those in IVF and B-B embryos.

Our previous studies have shown that the abnormal development of iSCTN embryos is caused by a failure to express genes related to ribosomal and mitochondrial functions [32,35]. Here, we selected IVF, B-B and O-B embryos at the 4-, 8-, and 16-cell stages, respectively, for transcriptome analysis and show that O-B embryos failed EGA during the 4–8 cell stages. GO enrichment analysis indicated that pathways with genes in mitochondrial transport, mitochondrial translational initiation, mitochondrial large ribosomal subunit and mitochondrial outer membrane were significantly enriched. In addition, pluripotent genes associated with ZGA [36] such as *Klf4*, *Otx1*, *Kdm5b*, *Tet1*, *Taf1d*, *Sp1*, *Nfya*, *Nfyb*, *Dppa4* and *Nanog* were not properly activated in O-B embryos.

The entry of mitochondrial proteins encoded by nuclear genes into mitochondria requires the involvement of the outer/inner membrane complex (TOM/TIM) embedded on the surface of mitochondria [37,38]. Mitochondrial targeting sequences within the precursor proteins are recognized by TOM20 and TOM70 receptors to guide the substrates into the TOM40 channel [39]. Interlinked sorting machines further direct proteins into mitochondrial sub-compartments upon translocation through the TOM40 channel [40]. In a previous report, we showed that the outer membrane/inner membrane family (Tom/Timm)-related genes had a significant downward trend in iSCNT embryo [32]. Most mitochondrial proteins are encoded by nuclear genes and are translated into precursors on cytoplasmic ribosomes prior to mitochondrial involvement. Mitochondrial ribosomes are responsible for the translations of 13 essential proteins during oxidative phosphorylation, and mitochondrial ribosomal proteins (MRPs) are components of mitochondrial ribosomes [41]. Abnormal MRPs can lead to a decrease in mitochondrial membrane potential and ATP production, causing various metabolic disorders. Studies have shown that MRPL48 is a key protein affecting mitochondrial function and plays a very important role in cell metabolism [42,43]. In this study, we found that the mRNA expressions of Tomm40 and Mrpl48 in O-B embryos were lower than those in IVF and B-B embryos. However, immunofluorescence staining indicated that the relative fluorescence intensity in O-B iSCNT embryos was higher than in B-B embryos, which may be a signal from maternal mitochondrial TOMM40 and MRPL48 proteins in mitochondria rather than a newly synthesized protein.

The structure and morphology of mitochondria changes significantly at different stages of development, which is controlled by mitochondrial fusion and fission processes and closely related to ATP supply or signal transduction [44,45]. *Mfn1* and *Mfn2* are the main genes regulating mitochondrial fusion, which regulate aerobic energy generation, ROS production, calcium homeostasis and cell signaling pathways [27]. When *Mfn1* expression was decreased, mitochondrial membrane division occurred, the morphology changed from a normal tubular junction network to a globule structure, and ATP synthesis decreased [46]. The lack of *Mfn1* or *Mfn2* resulted in mitochondrial fragmentation and incomplete tubular structure, leading to the abnormal fusion of mitochondria [28]. In yeast, blocking mitochondrial fusion results in defects in the respiratory chain [47]. Low expression of *Mfn1* was associated with mitochondrial dysfunction and led to reduced membrane potential and ATP levels in SCNT embryos and decreased female fertility [48,49]. In bovine SCNT embryo experiments, it has been confirmed that *Mfn1* could positively promote the development of cloned embryos [50]. In this study, we found that the low expression of *Mfn1* in O-B of iSCNT embryos is one of the important causes of EGA failure in O-B embryos. Overexpression or knockdown experiments demonstrated that *Mfn1* obviously affected mitochondrial fusion and, thus, affected mitochondrial functions, including ATP generation, mitochondrial membrane potential and ROS production. Moreover, overexpression of *Mfn1* induced O-B blastocyte development and promoted expressions of EGA-related genes. Therefore, the *Mfn1* gene may affect EGA by regulating mitochondrial biological functions in O-B iSCNT embryos.

In conclusion, the present study suggests that dysfunction of mitochondria-related genes is one of the main causes of EGA failure in O-B iSCNT embryos. *Mfn1* gene affects EGA and embryo development by regulating mitochondrial function. These findings provide new insights for our understanding of the mechanism by which *Mfn1* regulates embryonic development and EGA events.

## 4. Materials and Methods

### 4.1. Ethics Statement and Chemicals

All animal experimental procedures were performed in accordance with Ethical Approval for Research Involving Animals of Laboratory Animal Platform of Inner Mongolia University (IACUC Issue No. IMU-CATTLE-2021-041). Chemicals were purchased from Sigma Chemical Co. (St. Louis, MO, USA) unless otherwise mentioned.

### 4.2. Oocyte Collection and In Vitro Maturation

The ovaries were collected from the slaughterhouse, kept in a physiological saline solution containing penicillin and streptomycin at 20–25 °C, and transported to the laboratory within 2 h. The cumulus-oocyte complexes (COCs) were collected, followed by in vitro maturation according to our previous study [50]. Briefly, COCs were collected from 3- to 8 mm diameter antral follicles using a syringe needle, picked up and cleaned in washing buffer (M199 + 1% FBS) three times under a stereoscopic microscope. The COCs were then cultured in four-well plates (Nunc) in 500 µL of oocyte maturation media at 38.5 °C under 5% CO_2_. After 24 h of IVM, the bovine oocytes were transferred by pipette into 1-mg/mL hyaluronidase to remove cumulus cells. The first polar body extrusions were calculated as the maturation rate of the oocytes.

### 4.3. Donor Cell Preparation

Bovine and ovine fetal fibroblasts were isolated from 60-day fetuses from a ranch. Fetal fibroblasts were cultured in Dulbecco’s modified Eagle Medium (DMEM, Gibco, New York, NY, USA) containing 10 % fetal bovine serum (FBS, Gibco, New York, NY, USA), 1% MEM NEAA, 1% L-glutamine, and 1% penicillin-streptomycin under 5% CO_2_ in humidified air at 37 °C. Bovine and ovine fetal fibroblasts at passage 2–10 were used as donor cells for nuclear transfer.

### 4.4. Overexpression and Knockdown of Mfn1 Vector Construction

The target gene Mfn1 and the vector backbone pmCherry2-N1 (Addgene, Watertown, MA, USA) were digested and linearized with DNA restriction enzymes NotI and EcoRI. The target gene and the backbone vector were connected by T4 ligase to create the recombinant plasmid pmCherry2-N1-Mfn1. The DH5α competent cells were transformed, sequenced and compared successfully. Four short hairpin RNAs (shRNA) targeting the Mfn1 gene were designed, and the shRNAs were annealed to form double strands. The interference vector pGPU6-GFP-Neo was digested and recovered using restriction enzymes EcoRI and AgeI. The annealed diluted product was ligated with the linearized backbone vector Stbl3 E. Coli-competent cells were transformed and cultured on a plate overnight. Individual colonies were expanded for culture. PCR was performed for detection and PCR and Sanger sequence technology were used to screen and confirm the correct clone. Vector pGPU6-GFP-Neo-shMfn1 was obtained.

### 4.5. Transfection

Fetal fibroblasts were seeded into wells of a 24-well clear microplate. The constructed Mfn1 overexpression and knockdown vectors were transfected into bovine fetal fibroblasts using transfection reagents: Lipofectamine™ -LTX Reagent with PLUS™ Reagent (15338-100, Thermo, New York, NY, USA). The transfection mixture was added to the well containing the cell monolayer with 80% cell coverage. The plates were gently shaken and incubated for 6 h at 37 °C and 5% CO_2_. The medium was then replaced and the preparation was incubated for 48 h under the same conditions.

### 4.6. Somatic Cell Nuclear Transfer, Fusion and Activation of Reconstructed Embryos

This process refers to our previously published papers [51]; briefly, 20–30 selected oocytes were placed into a 30-µL micro-drop of M199 medium (Gibco, New York, NY, USA) containing 1% FBS and 7.5 µg/mL of cytochalasin B covered by mineral oil. Bovine oocytes were enucleated by aspirating the first polar body and the MII spindle plate. Successful enucleation was confirmed by staining with Hoechst 33342 and visualization under ultraviolet (UV) light. The reconstructed embryos were sandwiched with a pair of electrodes, and two electric DC pulses of 1.8  kV/cm were delivered for 20 μs each by a Voltain cell fusion system (Cryologic, Victoria, Australia). The fused SCNT embryos were activated by 5 µmol/L of ionomycin and cycloheximide (CHX). After that, the activated embryos were cultured in SOFaa medium containing 0.4% BSA for 48 h. The reconstructed embryos after cleavage were transferred into 40 μL drops of SOFaa containing 4% FBS until they developed to the blastocyst stage.

### 4.7. In Vitro Fertilization

The bovine in vitro fertilization process refers to our previously published paper [52]. Briefly, frozen bull semen straw was thawed at 37 °C for 30 s and blown into BO-SemenPrep solution (IVF Bioscience, Cornwall, UK). The semen solution was centrifugated at 600 g for 10 min. Subsequently, the sperm solution was resuspended to a suitable density (2–3 × 10^6^ sperm/mL); a 40 μL sperm dilution was added to 20–25 mature COCs in a droplet of 460 μL of BO-IVF medium (IVF Bioscience). After 20 h of IVF, the cumulus cells and redundant sperm were removed from the oocytes in a M199-based medium containing 0.1% bovine testicular hyaluronidase. The embryos were washed with BO-Wash (IVF Bioscience) and BO-IVC media, and the zygotes were cultured in the BO-IVC medium in a 95% humidified atmosphere with 5% CO_2_ at 38.5 °C. Half of the medium was replaced every two days.

### 4.8. Transcriptome Sequencing

Embryos derived from IVF, B-B SCNT and O-B iSCNT embryos were collected at the 4-cell, 8-cell and 16-cell stages. RNA fragments and libraries were established using 20 embryos/pool at each stage and were sequenced on an Illumina HiSeq 2000TM instrument (San Diego, CA, USA). A series of corrections and filtering processes were performed to reduce the impact of sequencing errors on assembly. We removed reads with N content greater than 5%, reads containing PolyA, low quality reads (number of mass bases less than 50%), linker contamination (reads with at least 10 bp that match the linker sequence with more than 2 mismatches), and duplicate impurity values such as repeats to obtain clean reads. BWA software was used to compare the clean reads to the reference genome. The software parameters used the default values. Bowtie software was used to compare the clean reads to the reference genes, and the RPM (RNA Seq by Expectation Maximization) tool was used to calculate the FPKM value of each transcript. The specific calculation formula is as follows:FPKM = (10^6 C)/〖NL/10〗^3 
FPKM is the expression level of the gene, C is the number of Fragments uniquely matched to the gene, N is the total number of Fragments uniquely matched to the reference gene, and L is the number of bases in the coding region of the gene. Differentially expressed genes were calculated using EdgeR, with a screening threshold of *p*-value < 0.05 and |log2 (foldchange)| > 1. GO enrichment analysis uses the DAVID database (The Database for Annotation, Visualization and Integrated Discovery). The default parameters were used for this analysis.

### 4.9. RNA Extraction and Quantitative Real-Time PCR

Total RNA was extracted from each pool of embryos (*n* = 3 pools of 20 embryos) or cells (Cells of a six well plate) using the Pico-Pure RNA Isolation Kit (Thermo). The reverse transcription–qPCR (RT-qPCR) was transcribed from total RNA using the SuperScript III First-Strand Synthesis System (Thermo) according to the manufacturer’s instructions. All primers were listed in Appendix A. The quantitative real-time PCR (qPCR) was performed using a TB Green Premix Ex Taq (Takara, Dalian, China), and the signals were detected with a ABI7500 Real-Time PCR System (Thermo). Analyses of the relative gene expression was measured using the 2^−ΔΔCt^ method.

### 4.10. Determinations of ROS Levels in 8-Cell Stage Embryos

The ROS levels of the oocytes and 8-cell stage embryos (20–30 embryos per group) were measured using the ROS assay kit (Beyotime, Shanghai, China). Briefly, the embryos were incubated in M199-based medium with 10 M 2’,7’-dichlorofluorescein diacetate at 38.5 °C for 20 min. The embryos were then washed in M199-based medium for 15 min at room temperature. After washout, the embryos were placed on glass slides and images were captured by a fluorescence microscope. ROS fluorescence intensity of images was quantified using ImageJ 1.45s software (National Institutes of Health, Bethesda, MD, USA).

### 4.11. Evaluation of Mitochondrial Distribution

Embryos (20 embryos per group) at the 8-cell stage were stained with 200 nmol/L Mito-tracker Red (Solarbio, Beijing, China) at 38.5 °C with 5% CO_2_ for 15 min. After being washed with PBS + 3%BSA three times, the embryos were fixed with 4% paraformaldehyde at room temperature for 15 min. The embryos were then washed three times with PBS + 3%BSA before being placed on slides and were observed by a laser confocal microscope (Nikon A1 Plus, Tokyo, Japan).

### 4.12. Mitochondrial Membrane Potential Analysis

Eight-cell embryos (20–25 embryos per group) were stained by JC-1, which is a fluorescent dye that aggregates in mitochondria and can show the membrane potential across the stromal membrane. Red fluorescence represents high membrane potential and green fluorescence represents low membrane potential. Mitochondrial activity was assessed by the ratio of red/green fluorescence intensity. Embryos at the 8-cell stage were washed three times in PBS-3% BSA, incubated in 2.5 µmol/L JC-1 (15003, Cayma Chemical, Ann Arbor, MI, USA) for 30 min, washed twice in PBS-3% BSA, and observed by confocal laser microscope. The green JC-1 signals were measured at 485/535 nm, whereas the red signals were measured at 590/610 nm. Image-J software was used for fluorescence intensity analysis.

### 4.13. ATP Level Detection

ATP levels in 8-cell-stage embryos (100 embryos per group) were detected using an enhanced ATP assay kit (Beyotime Biotechnology, Shanghai, China) in order to assess the mitochondrial function. The embryos were placed in 40 µL of ATP lysis buffer, and an appropriate amount of ATP detection working solution was prepared according to the number of samples tested. The ATP detection working solution was prepared with ATP detection reagent and ATP detection reagent diluent in a 1:4 ratio. After 100 µL of ATP detection working solution was added to each well of the 96-well plate and stood for 3 min at room temperature, 20 µL of sample or standard was added. The luminescence was measured using a multifunctional microplate reader with a detection time of 15 s. The concentration of ATP in the sample was calculated from the standard curve.

### 4.14. Immunofluorescence

Embryos were fixed in 4% paraformaldehyde overnight at 4 °C. The embryos (20–25 embryos per group) were then transferred to a membrane-permeable solution (0.1% Triton X-100) and incubated at room temperature for 1 h. After 1 h incubation in a blocking buffer of 3% BSA-PBS, embryos were subsequently treated with anti-Mfn1 antibody (Santa Cruz Biotechnology, SC-166644, diluted 1:500), anti-Mfn2 (Santa Cruz Biotechnology, Dalla, TX, USA, SC-515647, diluted 1:500), anti-TOMM40 antibody (Proteintech, Wuhan, China,18409-1-AP, diluted 1:500), anti-ATP1B3 antibody (Abeam, ab137055, diluted 1:1000) and MRPL48 (Proteintech, 14677-1- AP, diluted 1:500) overnight at 4 °C. After 5 washes with 0.3% BSA + PBS, oocytes were incubated with Alexa Fluor 488 secondary antibody (Proteintech, SA00013-2, diluted 1:1000, Santa Cruz Biotechnology, sc-516176, diluted 1:10). 1000) or Alexa Fluor 594 secondary antibody (Proteintech, SA00013-4, diluted 1:1000) for overnight incubation at 4 °C. Embryos were then placed on glass slides and examined with a confocal laser scanning microscope (Nikon A1 plus, Tokyo, Japan).

### 4.15. Western Blot Analysis

Harvested cells were washed with cold PBS, treated with 200 µL of protein lysate (990 µL RIPA + 10 µL PMSF) per 2 × 107 cells, and placed on ice overnight. After centrifugation at 4 °C, 12,000 rpm for 30 min, the supernatant was collected. Protein concentration was determined with Pierce BCA protein assay (Thermo Fisher Scientific). The protein supernatant and DTT-containing 4× protein loading buffer were mixed in a 3:1 ratio and heated in boiling water for 7 min to denature the protein. The mixtures were cooled and packed in a 20 μL/tube, and stored at −80 °C. Following the instructions in the SDS-PAGE gel preparation kit, 10% gel concentration was selected for electrophoresis. After electrophoresis, the mold was transferred and the PVDF membrane was blocked in 5% skim milk (2.5 g nonfat dry milk + 50 mL TBST) for 1 h at room temperature. The preparation was mixed with TBS-T (0.1% Tween) mouse anti-Mfn1 antibody (Santa Cruz Biotechnology, SC-166644, diluted 1:500) and mouse anti-GAPDH antibody (Proteintech, 60004-1-Ig, diluted 1:5000) and incubated overnight at 4 °C. After the primary antibody was aspirated and discarded, the membrane was washed with 1 × TBST washing solution with a shaker for 5 min 4 times. The membrane was then placed in 5 mL of 1 × TBST-diluted secondary antibody HRP-conjugated Affinipure Goat Anti-Mouse IgG (H + L) (Proteintech, SA00001-1, 1:5000) and incubated with shaking at room temperature for 1 h. After the secondary antibody was aspirated and discarded, the membrane was washed in 1×TBST washing solution on a shaker for 5 min for a total of 4 times. The membrane was then positioned face-up in a Ziploc bag. The developer solution was prepared with Luminol Reagent and Peroxide Solution in 1:1 ratio and 0.1 ml/cm^2^ of PVDF membrane. The developer solution was added slowly on the front of the film and covered with a Ziiploc bag without air bubbles. The film was exposed using a gel chemiluminescence system and the image was saved in the desired format. Image J software 1.8.0.112 (National Institutes of Health, Bethesda, MD, USA) was used to analyze the image to obtain gray value.

### 4.16. Statistical Analysis

The results were presented as the mean  ±  SD of three independent experiments. Statistical analyses were performed using the GraphPad Prism 8.3.0 software (San Diego, CA, USA). *p* < 0.05 was considered statistically significant.

## Figures and Tables

**Figure 1 ijms-23-10145-f001:**
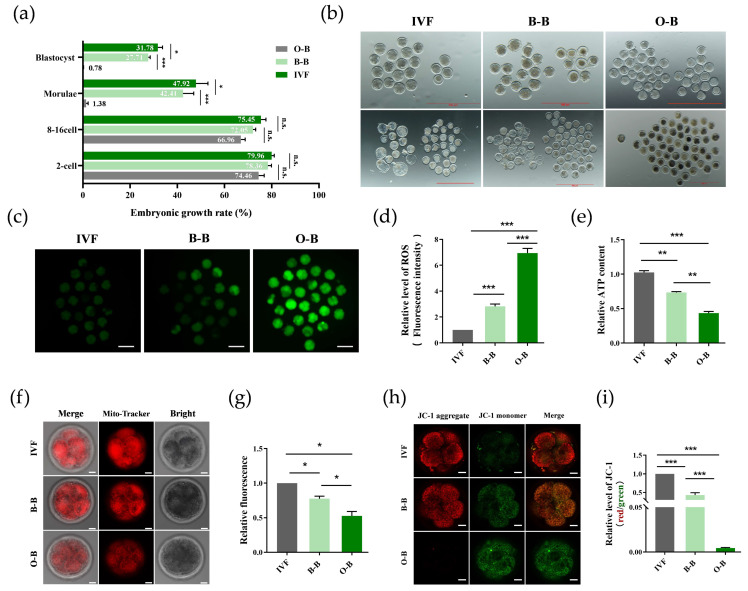
The development efficiency of IVF, B-B and O-B embryo and mitochondrial function test. (**a**), The development efficiency of IVF, B-B and O-B embryo analysis, respectively; (**b**), Representative images of the 8-cell and blastocytes of IVF, B-B and O-B embryos, respectively, scale bars: 500 μm.(**c**,**d**), Quantification of ROS level (green color) in 8-cell stage embryos (scale bars: 100 μm)and the relative intensities analysis of ROS signal. (**e**), The relative ATP content analysis in different embryos. (**f**,**g**), Representative images of MitoTracker-stained embryos (red color scale bars: 20 μm), and the relative fluorescence analysis of embryos. (**h**,**i**) Mitochondrial membrane potential analysis of 8-cell IVF, B-B and O-B embryos by JC-1 staining, and relative quantification of J-aggregate(red)/J-monomer(green) signal in different embryos, scale bars: 20 μm.( * stands for *p* < 0.05, ** stands for *p* < 0.01, *** stands for *p* < 0.001).

**Figure 2 ijms-23-10145-f002:**
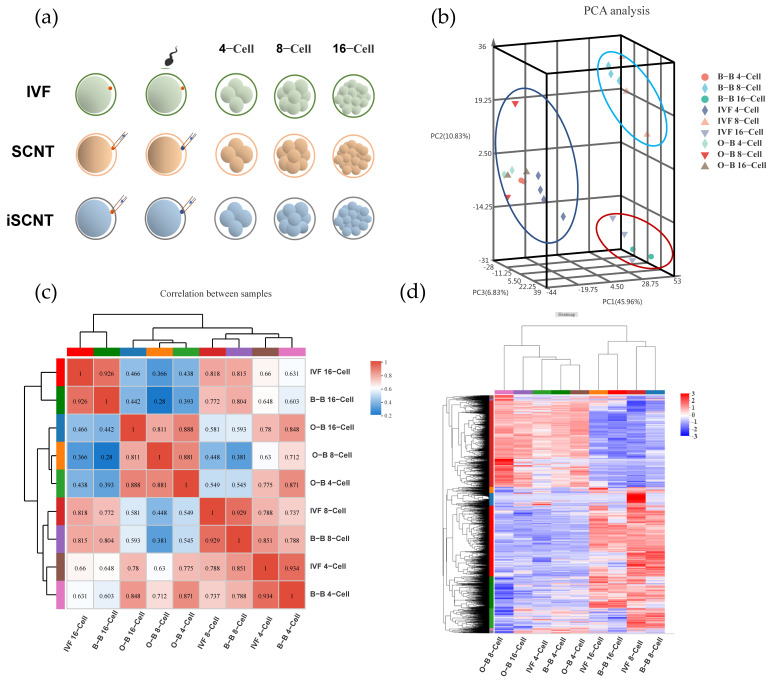
RNA-Sequencing analysis of different embryos. (**a**), schematic diagram of sample collection; (**b**), PCA analysis between samples; (**c**), correlation analysis between samples; (**d**), heatmap analysis of gene expression between samples.

**Figure 3 ijms-23-10145-f003:**
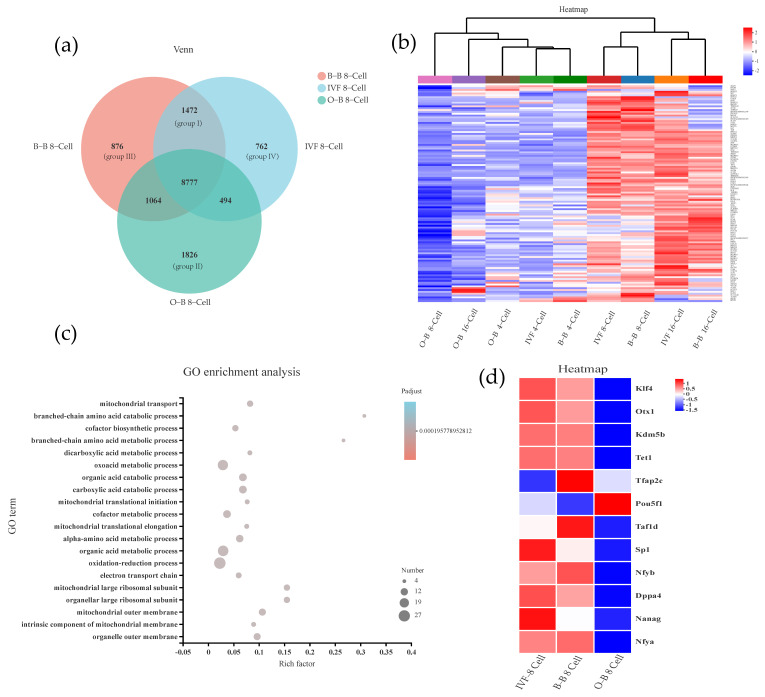
Differentially expressed genes and GO analysis of differential embryos. (**a**) Venn diagram analysis of 8-cell stage embryos of IVF, B-B and O-B embryos; (**b**) Heat map analysis of mitochondrial-related genes; (**c**) GO enrichment analysis; (**d**) ZGA-related pluripotency genes analysis of different embryos.

**Figure 4 ijms-23-10145-f004:**
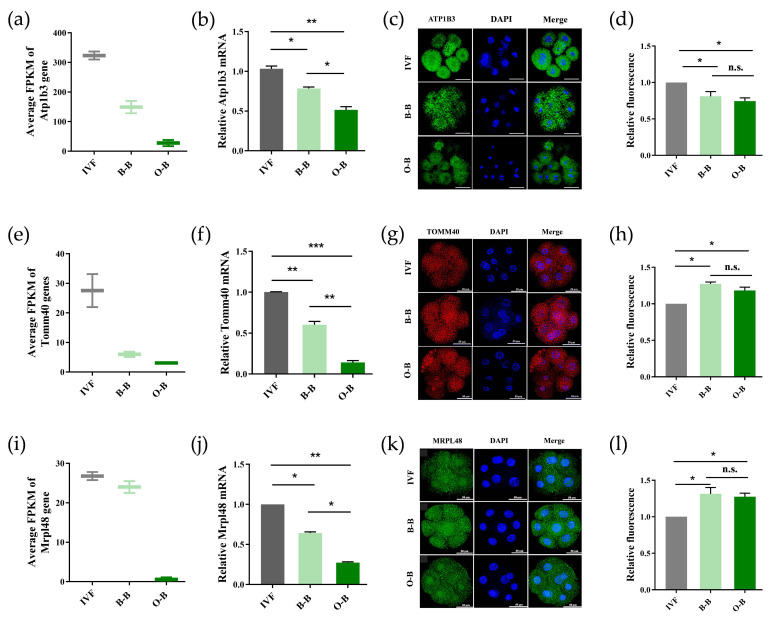
Expression analysis of mitochondrial function related genes in IVF, SCNT and iSCNT 8-cell stage embryos. Transcriptome (**a**), RT-qPCR expression (**b**), immunofluorescence staining (green) (**c**) and the fluorescence intensity profiles (**d**) analysis of *Atp1b3* gene in IVF, B-B and O-B SCNT embryos at 8-cell stages; Transcriptome (**e**), RT-qPCR expression (**f**), immunofluorescence staining (red) (**g**) and the fluorescence intensity profiles (**h**) analysis of *TOMM40* gene in IVF, B-B and O-B SCNT embryos at 8-cell stages; Transcriptome (**i**), RT-qPCR expression (**j**), immunofluorescence staining (green) (**k**) and the fluorescence intensity profile s (**l**) analysis of *MRPL48* gene in IVF, B-B and O-B SCNT embryos at 8-cell stages. (* stands for *p* < 0.05, ** stands for *p* < 0.01, *** stands for *p* < 0.001, scale bars: 50 μm).

**Figure 5 ijms-23-10145-f005:**
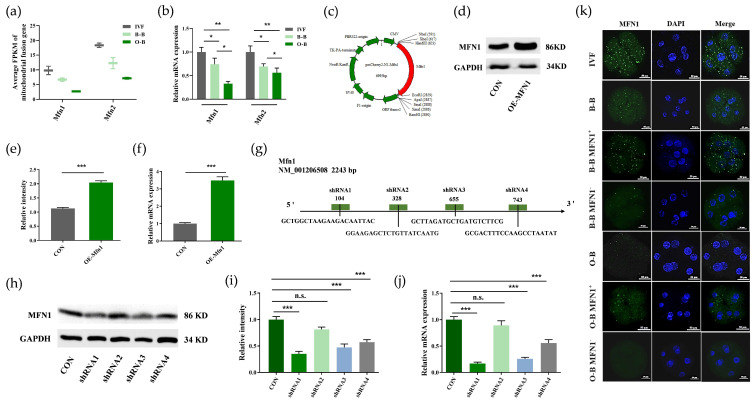
Expression analysis of mitochondrial fusion genes in embryos, and construction and function verification of overexpressed and knockdown *Mfn1* gene vectors. (**a**) RNA-SEQ data analysis of *Mfn1* and *Mfn2* in IVF, B-B and O-B 8-cell embryos; (**b**) RT-qPCR verification of *Mfn1* and *Mfn2* in IVF, B-B and O-B 8-cell embryos; (**c**) Sketch of overexpressed *Mfn1* gene vector containing a CMV promoter and kanamycin tag. (**d**,**e**) Western blot and fluorescence intensity profiles analysis of OE- Mfn1 cells; (**f**) RT-qPCR verification of OE- Mfn1 cells; G, Schematic of *Mfn1* loci and diagram of shRNA targeting sites; (**h**,**i**) Western blot and the fluorescence intensity profiles analysis of knockdown *Mfn1* cells. (**j**) RT-qPCR verification of knockdown *Mfn1* cells; (**k**) immunofluorescence staining of MFN1 (green) in different embryos, scale bar: 50 μm. (* stands for *p* < 0.05, ** stands for *p* < 0.01, *** stands for *p* < 0.001).

**Figure 6 ijms-23-10145-f006:**
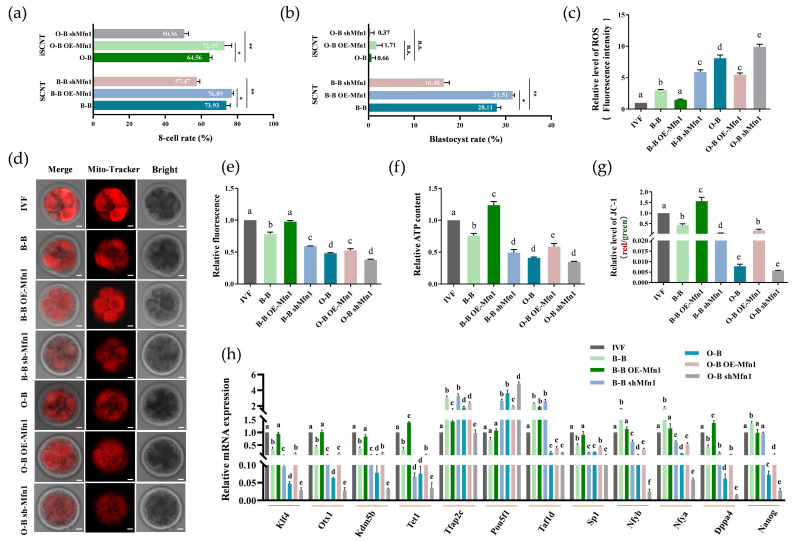
Embryo development, mitochondrial function and pluripotent gene detection in Mfn1 gene overexpressed or knockdown in B-B and O-B embryos. (**a**) Comparative analysis of the 8-cell development rate of overexpression (OE-Mfn1) or knockdown (shMfn1) of Mfn1 gene in B-B and O-B embryos (* stands for *p* < 0.05, ** stands for *p* < 0.01); (**b**) Blastocyst development rate of Mfn1 gene overexpressed or knockdown B-B and O-B embryos (n.s., not significant difference, * stands for *p* < 0.05, ** stands for *p* < 0.01). (**c**) Relative ROS levels analysis of 8-cell stage different embryos; (**d**,**e**) Representative image of MitoTracker-stained (red) 8-cell Mfn1 gene overexpressed or knockdown B-B and O-B embryos, and the relative fluorescence analysis of different embryos; (**f**) Relative ATP content analysis in different embryo groups; (**g**) Relative quantification of J-aggregate/ J-monomer signal in 8-cell Mfn1 gene overexpressed or knockdown B-B and O-B embryos. (**h**) ZGA-related pluripotency genes analysis of 8-cell Mfn1 gene overexpressed or knockdown B-B and O-B embryos. Different letters represent significant differences. scale bars: 20 μm.

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
