# Peer review of "Low Expression of Mitofusin 1 Gene Leads to Mitochondrial Dysfunction and Embryonic Genome Activation Failure in Ovine-Bovine Inter-Species Cloned Embryos"

_ijms, 2022, doi:10.3390/ijms231710145_

Round 1

Reviewer 1 Report

The authors showed the importance of mitochondrial energy and their difference based on the generation methods. The study is well designed, however, the ethical problem is a huge problem. This study focused on the generation of the inter-species embryos and this is scientifically needed at some point, however, it is not an ethical not authorized detail. And the interpretation of the results has a bias on the difference in the species.

The results should be demonstrated in the same species with different methods of embryo generation.  

Author Response

Response 1:

Thanks to the reviewer for your questions, the question raised by the reviewer is very worthy of attention. We also took this into consideration when designing our experiments. In interspecific somatic cell nuclear transfer (iSCNT), donor cells are transplanted into a recipient enucleated oocyte of a different species/family/order/class to construct a heterogeneous reconstructed embryo. ISCNT technology is potentially significant for rescue of endangered animals and therapeutic cloning [1-3] . There have been many research reports in this field of study, especially in the study of the relationship between nucleus and cytoplasm in early embryo, iSCNTcan provide a model of embryonic genome activation(EGA) events[4-10]. The purpose of our research on bovine-ovine iSCNT embryos is to solve the problem of low efficiency encountered in bovine somatic nuclear transfer, and only the early embryo stage is achieved.The experimental process is conducted according to the Ethical Approval for Research Involving Animals of Laboratory Animal Platform of Inner Mongolia University(IACUC Issue No. IMU-CATTLE-2021-041). We have supplemented it in the materials and methods of the revised manuscript-clean (Line 330-333).For details, see the following attachments:

References

  1. Loi, P.; Galli, C.; Ptak, G., Cloning of endangered mammalian species: any progress? Trends in biotechnology 2007, 25, (5), 195-200.
  2. Lagutina, I.; Fulka, H.; Lazzari, G.; Galli, C., Interspecies somatic cell nuclear transfer: advancements and problems. Cellular reprogramming 2013, 15, (5), 374-84.
  3. Matoba, S.; Zhang, Y., Somatic Cell Nuclear Transfer Reprogramming: Mechanisms and Applications. Cell stem cell 2018, 23, (4), 471-485.
  4. Lagutina, I.; Fulka, H.; Brevini, T. A.; Antonini, S.; Brunetti, D.; Colleoni, S.; Gandolfi, F.; Lazzari, G.; Fulka, J., Jr.; Galli, C., Development, embryonic genome activity and mitochondrial characteristics of bovine-pig inter-family nuclear transfer embryos. Reproduction 2010, 140, (2), 273-85.
  5. Lagutina, I.; Zakhartchenko, V.; Fulka, H.; Colleoni, S.; Wolf, E.; Fulka, J., Jr.; Lazzari, G.; Galli, C., Formation of nucleoli in interspecies nuclear transfer embryos derived from bovine, porcine, and rabbit oocytes and nuclear donor cells of various species. Reproduction 2011, 141, (4), 453-65.
  6. Kwon, D.; Koo, O. J.; Kim, M. J.; Jang, G.; Lee, B. C., Nuclear-mitochondrial incompatibility in interorder rhesus monkey-cow embryos derived from somatic cell nuclear transfer. Primates; journal of primatology 2016, 57, (4), 471-8.
  7. Jeon, Y.; Nam, Y. H.; Cheong, S. A.; Kwak, S. S.; Lee, E.; Hyun, S. H., Absence of nucleolus formation in raccoon dog-porcine interspecies somatic cell nuclear transfer embryos results in embryonic developmental failure. The Journal of reproduction and development 2016, 62, (4), 345-50.
  8. Zuo, Y.; Gao, Y.; Su, G.; Bai, C.; Wei, Z.; Liu, K.; Li, Q.; Bou, S.; Li, G., Irregular transcriptome reprogramming probably causes thec developmental failure of embryos produced by interspecies somatic cell nuclear transfer between the Przewalski's gazelle and the bovine. BMC genomics 2014, 15, 1113.
  9. Hua, S.; Lu, C.; Song, Y.; Li, R.; Liu, X.; Quan, F.; Wang, Y.; Liu, J.; Su, F.; Zhang, Y., High levels of mitochondrial heteroplasmy modify the development of ovine-bovine interspecies nuclear transferred embryos. Reproduction, fertility, and development 2012, 24, (3), 501-9.
  10. Jiang, Y.; Kelly, R.; Peters, A.; Fulka, H.; Dickinson, A.; Mitchell, D. A.; St John, J. C., Interspecies somatic cell nuclear transfer is dependent on compatible mitochondrial DNA and reprogramming factors. Plos One 2011, 6, (4), e14805.

Reviewer 2 Report

This manuscript prepared well and designed scientifically. Therefore, my recommendation for this manuscript is acceptance after minor revision. Thanks.

Author Response

Response 1:

Thank the reviewer for your recognition of this research. Your encouragement will be a great impetus for my further research.

Reviewer 3 Report

Overview

Authors used ovine-to-bovine SCNT as a model of interspecies nuclear transfer and compared the development of these embryos with bovine-to-bovine SCNT and IVF bovine embryos. Authors found that ovine-to-bovine SCNT embryos are developmentally arrested at 8-16 cell stage and do not develop into morula/blastocysts.

At the molecular level, mitochondrial homeostasis was significantly impaired in ovine-to-bovine 8-cell embryos as judged by increased ROS, decreased ATP levels and mitochondrial membrane potential. At the gene expression level RNA-seq analysis showed that in ovine-to-bovine embryo genome activation did not occur. Expression levels of mitochondrial and pluripotency genes were significantly lower in o-t-b embryos.

Major issues

Authors conclude that the observed defects (EGA failure etc.) are caused by abnormal expression of mitochondrial related genes (lines 183,184, 323,324). But in my opinion, the presented data do not allow to draw such conclusion. In my opinion, it could be possible that ovine-to-bovine embryos fail to activate nuclear genome for some unknown reason, thus impairing expression of mitochondrial genes, and this is one of the reasons that leads to developmental arrest. Mfn1 rescue experiments confirm that proper functioning of mitochondria is important for normal embryo development.

So, from the presented data it not possible to conclude if EGA failure leads to mitochondrial dysfunction and developmental arrest, or vice versa, mitochondrial dysfunction occurs first, then embryos fail to activate gene transcription program.

Specific points

Fig.1 and material and methods section lack information on how many embryos were assayed per each condition.

The same notion is applied for Fig.4.

Fig.5 – It should be clarified in the text what is OE-Mfn cells. Is it embryonic fibroblasts? In the text it should be clearly stated which results were obtained on transfected cells and which on embryos.

Fig.5k apparently shows embryos with OE/KD of Mfn1, but the description of these embryos is related to Fig.6.

The text of manuscript describing Fig.5 should be thoroughly corrected.

In addition, results on Mfn1 expression from Fig.5 could be combined with the results on other genes from Fig.4 since they are presented in a similar way. Fig.5 should be dedicated to development of Mfn1 knock-down and overexpression system.

Author Response

Point 1: 
Overview
Authors used ovine-to-bovine SCNT as a model of interspecies nuclear transfer and compared the development of these embryos with bovine-to-bovine SCNT and IVF bovine embryos. Authors found that ovine-to-bovine SCNT embryos are developmentally arrested at 8-16 cell stage and do not develop into morula/blastocysts.
At the molecular level, mitochondrial homeostasis was significantly impaired in ovine-to-bovine 8-cell embryos as judged by increased ROS, decreased ATP levels and mitochondrial membrane potential. At the gene expression level RNA-seq analysis showed that in ovine-to-bovine embryo genome activation did not occur. Expression levels of mitochondrial and pluripotency genes were significantly lower in o-t-b embryos.
Response 1: We greatly appreciate the reviewer valuable comments regarding our manuscript. Thanks for your clearness and conciseness summary.

Major issues
Point 2: Authors conclude that the observed defects (EGA failure etc.) are caused by abnormal expression of mitochondrial related genes (lines 183,184, 323,324). But in my opinion, the presented data do not allow to draw such conclusion. In my opinion, it could be possible that ovine-to-bovine embryos fail to activate nuclear genome for some unknown reason, thus impairing expression of mitochondrial genes, and this is one of the reasons that leads to developmental arrest. Mfn1 rescue experiments confirm that proper functioning of mitochondria is important for normal embryo development.
So, from the presented data it not possible to conclude if EGA failure leads to mitochondrial dysfunction and developmental arrest, or vice versa, mitochondrial dysfunction occurs first, then embryos fail to activate gene transcription program.
 Response 2:  We thank the Reviewer for these positive comments. the conclusions are grossly overstated, for which we apologize. Our and others previous related studies have shown that there were many reasons for the failure of EGA in iSCNT embryos, including abnormal epigenetic modification, abnormal expression of mitochondrial realted genes, and failure of nucleolar remodeling[1-5]. Mitochondria also fulfil a wide range of functions beyond ATP synthesis, including the production of intracellular reactive oxygen species and calcium regulation, and are active participants in the regulation of signal transduction pathways. It encodes 13 subunits of the electron transfer chain (ETC), while nuclear DNA encodes approximately 1500 proteins involved in mitochondrial functions, which explains why interactions between the nucleus and mitochondria are absolutely necessary for proper mitochondria genesis and function[6-9]. In SCNT embryos, the relationship between the regulation of nucleus and mitochondria has always been a focus of research. Mitochondria undergo morphological changes in the early embryonic stage which regulated by the Mfn1 gene, and this morphological change would lead to changes in mitochondrial metabolic capacity. The metabolites affect the epigenetic modification of chromosomes to regulate gene expression[10, 11]. On this basis, this may explain why mitochondrial dysfunction affect embryonic developmental and EGA. We have revised the conclusions presented by the reviewers (lines 183, 184, 323, 324) in the revised manuscript-clean lines 179-181 323-327.

Specific points
Point 3: Fig.1 and material and methods section lack information on how many embryos were assayed per each condition.
 Response 3: Thank the reviewer for your suggestion. We accept the reviewer's suggestion. We have supplemented the number of embryos in the embryonic development(Table S1 and S2) and other conditional assays(in the material and methods section).

Table S1. The development efficiency of IVF, B-B and O-B embryo analysis, respectively
Groups Reconstructed embryos 2-cell 8-16 cell Morulae Blastocyst
IVF 453 362 (79.96) a 341 (75.45) a 216 (47.92) a 144 (31.78) a
B-B 415 325 (78.36)a 299 (72.05)a 176 (42.41)b 115 (27.71)b
O-B 387 288 (74.46)a 259 (66.96)a 5 (1.38)c 3 (0.78)c
Note: Values with different letters within the same column differ significantly (P<0.05).

Table S2. Comparative analysis of the 8-cell and blastocyst development rate of overexpression (OE-Mfn1) or knockdown (shMfn1) of Mfn1 gene in B-B and O-B embryos
Groups Reconstructed embryos 8-cell Blastocyst
SCNT B-B  249 184 (73.93) a 70 (28.11)a
 B-B OE-Mfn1 238 183 (76.89)b 75 (31.51)b
 B-B shMfn1 261 150 (57.47)c 43 (16.48)c
iSCNT O-B  302 195 (64.56)a 2 (0.66)a
 O-B OE-Mfn1 292 213 (72.95)b 5 (1.71)b
 O-B shMfn1 267 135 (50.56)c 1 (0.37)c
Note: Values with different letters within the same column differ significantly (P<0.05).

Point 4:The same notion is applied for Fig.4.
Response 4: Thank the reviewer for your suggestion.We accept the reviewer's suggestion. We have supplemented the number of embryos assays in the material and methods section.

Point 5:Fig.5 – It should be clarified in the text what is OE-Mfn cells. Is it embryonic fibroblasts? In the text it should be clearly stated which results were obtained on transfected cells and which on embryos.
Response 5: Thank the reviewer for your suggestion. We accept the reviewer's suggestion. We have described each cell and embryo in the revised manuscript-clean (shown in the Line-199-214).

Point 6:Fig.5k apparently shows embryos with OE/KD of Mfn1, but the description of these embryos is related to Fig.6.
Response 6: Thank the reviewer for your suggestion. Thank you for giving our manuscript a lot of consideration. We also tried to rearrange the order of the figures as you suggested. However, we found that the current way of expressing could better highlight the ideas we expressed. Fig5K was a functional validation on embryos after overexpression and knockdown of Mfn1, and we found that OE/KD vectors exhibited their effects at both the cellular and embryonic levels. Therefore, we still think that it is more clear to the reader to express in Fig5. 

Point 7:The text of manuscript describing Fig.5 should be thoroughly corrected.
Response 7: Thank the reviewer for your suggestion. We accept the reviewer's suggestion. Wehave thoroughly corrected the describtion of Fig.5 in the revised manuscript ( as shown in the revised manuscript-clean Line 199-214).

Point 8:In addition, results on Mfn1 expression from Fig.5 could be combined with the results on other genes from Fig.4 since they are presented in a similar way. Fig.5 should be dedicated to development of Mfn1 knock-down and overexpression system.
Response 8: Thank the reviewer for your suggestion.Thank you for giving our manuscript a lot of consideration. We also tried to rearrange the order of the figures as you suggested. However, we found that the current way of expressing could better highlight the ideas we expressed. In Fig. 4, we shown the mRNA and proteins level validation of genes which affecting mitochondrial function in different embryos; In Fig. 5, we put our focus on the Mfn1 gene and functional verification.Therefore, we chose to display the expression of Mfn1 gene in the set of Fig5, which is more convenient for readers to understand.
References
1. Zuo, Y. C.; Su, G. H.; Cheng, L.; Liu, K.; Feng, Y.; Wei, Z. Y.; Bai, C. L.; Cao, G. F.; Li, G. P., Coexpression analysis identifies nuclear reprogramming barriers of somatic cell nuclear transfer embryos. Oncotarget 2017, 8, (39), 65847-65859.
2. Zuo, Y.; Gao, Y.; Su, G.; Bai, C.; Wei, Z.; Liu, K.; Li, Q.; Bou, S.; Li, G., Irregular transcriptome reprogramming probably causes thec developmental failure of embryos produced by interspecies somatic cell nuclear transfer between the Przewalski's gazelle and the bovine. BMC genomics 2014, 15, 1113.
3. Su, G.; Wang, L.; Gao, G.; Wu, S.; Yang, L.; Wu, M.; Liu, X.; Yang, M.; Wei, Z.; Bai, C.; Li, G., C23 gene regulates the nucleolin structure and biosynthesis of ribosomes in bovine intraspecific and interspecific somatic cell nuclear transfer embryos. FASEB journal : official publication of the Federation of American Societies for Experimental Biology 2021, 35, (11), e21993.
4. Van Blerkom, J., Mitochondria in early mammalian development. Seminars in cell & developmental biology 2009, 20, (3), 354-64.
5. Lima, A.; Burgstaller, J.; Sanchez-Nieto, J. M.; Rodriguez, T. A., The Mitochondria and the Regulation of Cell Fitness During Early Mammalian Development. Current topics in developmental biology 2018, 128, 339-363.
6. Wallace, D. C.; Chalkia, D., Mitochondrial DNA genetics and the heteroplasmy conundrum in evolution and disease. Cold Spring Harbor perspectives in biology 2013, 5, (11), a021220.
7. Friedman, J. R.; Nunnari, J., Mitochondrial form and function. Nature 2014, 505, (7483), 335-43.
8. Amigo, I.; da Cunha, F. M.; Forni, M. F.; Garcia-Neto, W.; Kakimoto, P. A.; Luevano-Martinez, L. A.; Macedo, F.; Menezes-Filho, S. L.; Peloggia, J.; Kowaltowski, A. J., Mitochondrial form, function and signalling in aging. The Biochemical journal 2016, 473, (20), 3421-3449.
9. Kelly, D. P.; Scarpulla, R. C., Transcriptional regulatory circuits controlling mitochondrial biogenesis and function. Genes & development 2004, 18, (4), 357-68.
10. Zhang, J.; Zhao, J.; Dahan, P.; Lu, V.; Zhang, C.; Li, H.; Teitell, M. A., Metabolism in Pluripotent Stem Cells and Early Mammalian Development. Cell metabolism 2018, 27, (2), 332-338.
11. Nagaraj, R.; Sharpley, M. S.; Chi, F.; Braas, D.; Zhou, Y.; Kim, R.; Clark, A. T.; Banerjee, U., Nuclear Localization of Mitochondrial TCA Cycle Enzymes as a Critical Step in Mammalian Zygotic Genome Activation. Cell 2017, 168, (1-2), 210-223 e11.

Reviewer 4 Report

In this manuscript, Wu et al have shown the importance of mitochondrial function for a successful ovine-bovine interspecies embryo formation. They have identified Mitofusin gene as a key component in this process. Using in vitro systems to either downregulate or over-express the Mitofusin gene, they have confirmed the regulatory role of Mfn1 in the formation of O-B embryos. The experimental evidence is clear enough to validate their claim on Mfn1. However, a few questions need to be answered concerning the biological significance of this finding. In my opinion, the following points need to be addressed before the manuscript can go to publication.

1. Mfn1 is primarily involved in mitochondrial fusion. Although the involvement of Mfn1 is clear enough, the question remains as to which is the primary checkpoint - mitochondrial function or mitochondrial morphology. In other words, is Mfn1 the primary player or an associate to some other molecule? It would be interesting if mitochondrial function could be restored by some alternate means, like supplementation with ATP or antioxidants.

2. Since both Mfn1 and Mfn2 genes were downregulated, it would be more appropriate if both Mfn1 and Mfn2 genes were targetted for the overexpression experiments. 

Author Response

Point 1:.In this manuscript, Wu et al have shown the importance of mitochondrial function for a successful ovine-bovine interspecies embryo formation. They have identified Mitofusin gene as a key component in this process. Using in vitro systems to either downregulate or over-express the Mitofusin gene, they have confirmed the regulatory role of Mfn1 in the formation of O-B embryos. The experimental evidence is clear enough to validate their claim on Mfn1. However, a few questions need to be answered concerning the biological significance of this finding. In my opinion, the following points need to be addressed before the manuscript can go to publication.

Response 1: We greatly appreciate the Reviewer valuable comments regarding our manuscript. Thanks for your clearness and conciseness summary.

Point 2:. Mfn1 is primarily involved in mitochondrial fusion. Although the involvement of Mfn1 is clear enough, the question remains as to which is the primary checkpoint – mitochondrial function or mitochondrial morphology. In other words, is Mfn1 the primary player or an associate to some other molecule? It would be interesting if mitochondrial function could be restored by some alternate means, like supplementation with ATP or antioxidants.

Response 2: We thank the Reviewer for these valuable comments. As the Reviewer pointed out, the checkpoint involved in the regulation of mitochondrial fusion are: mitofusin 1 and 2 (Mfn1 and Mfn2) and OPA1. Mitochondrial fusion is a two step process, where the outer and inner mitochondrial membranes fuse by separate events  which can be explained by the mitochondrial sublocalization of Mfn1, Mfn2 and OPA1. Mfn1 and Mfn2 are located in the outer mitochondrial membrane, where they can regulate the outer membrane fusion, and OPA1 is located in the inner mitochondrial membrane, where it can regulate the inner membrane fusion[1, 2]. In bovine SCNT embryo experiments, it has been confirmed that Mfn1 can positively promote the development of cloned embryos[3]. Therefore, we focused our study on the regulation of Mfn1. Most recently, it was found that another molecular protein, Miro1, cooperates with mitochondrial fusion protein, as an inhibitory regulator of MFN at elevated mitochondrial Ca2+ levels[4]. This provides a good reference for us to further study the mechanism of mitochondrial fusion proteins. As the Reviewer pointed out, if mitochondrial function could be restored by some alternate means, like supplementation with ATP or antioxidants? This is indeed a good question. As far as we know, mitochondria undergo morphological changes in the early embryonic stage which regulated by the Mfn1 gene, and this morphological change would lead to changes in mitochondrial metabolic capacity. The metabolites affect the epigenetic modification of chromosomes to regulate gene expression[5, 6]. Therefore, it is an interesting question whether the additive will change the function of mitochondria and then regulate the expression of genes, and it is worth exploring in our follow-up experimental studies.

Point 3: Since both Mfn1 and Mfn2 genes were downregulated, it would be more appropriate if both Mfn1 and Mfn2 genes were targetted for the overexpression experiments.

Response 3: We thank the reviewer for these valuable comments. As the reviewer pointed out, qPCR ananlysis show that both Mfn1 and Mfn2 genes were downregulated in O-B iSCNT embryos in our experiment. However, we found that the Mfn2 protein was not significantly different between B-B and O-B embryos by immunofluorescence assays (shown in Figure S1), so we focused on the Mfn1 gene in the further .study of our manuscript.The reviewer's suggestion provides a good idea for our future reseach.

Figure S1. The immunofluorescence staining(A) and the fluorescence intensity profiles analysis(B)of Mfn1 and Mfn2 gene in IVF, B-B and O-B SCNT embryos at 8-cell stages.

References

  1. Zorzano, A.; Liesa, M.; Sebastian, D.; Segales, J.; Palacin, M., Mitochondrial fusion proteins: dual regulators of morphology and metabolism. Seminars in cell & developmental biology 2010, 21, (6), 566-74.
  2. Malka, F.; Guillery, O.; Cifuentes-Diaz, C.; Guillou, E.; Belenguer, P.; Lombes, A.; Rojo, M., Separate fusion of outer and inner mitochondrial membranes. EMBO reports 2005, 6, (9), 853-9.
  3. Hua, S.; Zhang, H.; Song, Y. K.; Li, R. Z.; Liu, J.; Wang, Y. S.; Quan, F. S.; Zhang, Y., High expression of Mfn1 promotes early development of bovine SCNT embryos: Improvement of mitochondrial membrane potential and oxidative metabolism. Mitochondrion 2012, 12, (2), 320-327.
  4. Fatiga, F. F.; Wang, L. J.; Hsu, T.; Capuno, J. I.; Fu, C. Y., Miro1 functions as an inhibitory regulator of MFN at elevated mitochondrial Ca2+ levels. J Cell Biochem 2021, 122, (12), 1848-1862.
  5. Zhang, J.; Zhao, J.; Dahan, P.; Lu, V.; Zhang, C.; Li, H.; Teitell, M. A., Metabolism in Pluripotent Stem Cells and Early Mammalian Development. Cell metabolism 2018, 27, (2), 332-338.
  6. Nagaraj, R.; Sharpley, M. S.; Chi, F.; Braas, D.; Zhou, Y.; Kim, R.; Clark, A. T.; Banerjee, U., Nuclear Localization of Mitochondrial TCA Cycle Enzymes as a Critical Step in Mammalian Zygotic Genome Activation. Cell 2017, 168, (1-2), 210-223 e11.

Round 2

Reviewer 1 Report

A revised manuscript has fulfilled the question of the reviewer. now it is suitable for publication.